# Does Vitamin D Supplementation Slow Brain Volume Loss in Multiple Sclerosis? A 4-Year Observational Study

**DOI:** 10.3390/nu17142271

**Published:** 2025-07-09

**Authors:** Weronika Galus, Mateusz Winder, Aleksander J. Owczarek, Anna Walawska-Hrycek, Michalina Rzepka, Aleksandra Kaczmarczyk, Joanna Siuda, Ewa Krzystanek

**Affiliations:** 1Department of Neurology, Faculty of Medical Sciences in Katowice, Medical University of Silesia, 40-752 Katowice, Poland; awalawska-hrycek@sum.edu.pl (A.W.-H.); michalinaj93@gmail.com (M.R.); akaczmarczyk@sum.edu.pl (A.K.); jsiuda@sum.edu.pl (J.S.); 2Department of Radiology and Nuclear Medicine, Faculty of Medical Sciences in Katowice, Medical University of Silesia, 40-752 Katowice, Poland; mwinder@sum.edu.pl; 3Department of Pathophysiology, Faculty of Medical Sciences in Katowice, Medical University of Silesia, 40-752 Katowice, Poland; aowczarek@sum.edu.pl; 4Department of Neurology, Faculty of Health Sciences in Katowice, Medical University of Silesia, 40-635 Katowice, Poland; ekrzystanek@sum.edu.pl

**Keywords:** multiple sclerosis, magnetic resonance imaging, vitamin D, supplementation

## Abstract

**Background and Aims:** Vitamin D is currently well regarded for its pleiotropic effects on the immune system, stimulating an anti-inflammatory response and enhancing immune tolerance. Vitamin D deficiency is an established risk factor for multiple sclerosis (MS). Additionally, lower vitamin D serum levels are associated with worse disease outcomes. However, current randomized clinical trials provide conflicting evidence about the beneficial role of vitamin D on disease progression. Most studies have evaluated the effect of vitamin D supplementation on clinical and radiological activity, yet very few have examined the impact on brain atrophy. **Methods:** A 4-year observational, non-interventional study design was applied to evaluate the association between vitamin D supplementation and disease progression. Altogether, 132 relapsing–remitting multiple sclerosis patients were enrolled in the study (97 subjects in the group with vitamin D supplementation and 35 subjects in the group without supplementation). The analyzed groups were similar in terms of age, body mass index, sun exposure, comorbidities, nicotinism, duration of the disease, and current treatment. The number of relapses, Expanded Disability Status Scale assessments, and the number of new/enlarged T2-weighted lesions and gadolinium-enhancing lesions in magnetic resonance imagining analyses, as well as 25-hydroxyvitamin D serum levels, were assessed every 12 months of a 4-year follow-up, whereas brain atrophy was assessed at the baseline and after 36 months using two-dimensional measurements. **Results:** After 36 months, a significant increase in atrophy was observed in both groups; however, patients without vitamin D supplementation had a significantly higher increase in intercaudate distance, third ventricle width, and bicaudate ratio after 36 months of observation (*p* < 0.05). Vitamin D supplementation among the studied group did not affect other disease activity outcomes. **Conclusions:** Our study revealed an observed association between vitamin D supplementation and reduced brain atrophy in patients with MS. Randomized controlled trials are required to establish the impact of vitamin D supplementation on brain atrophy progression.

## 1. Introduction

Vitamin D has recently gained attention for its role in stimulating anti-inflammatory pathways and supporting immune tolerance [1]. An optimal vitamin D status may have a protective role against the development of autoimmune diseases [2], including multiple sclerosis (MS). MS is characterized as an immune-mediated disorder of the central nervous system (CNS) and is experiencing a growing global prevalence [3]. Inflammatory and neurodegenerative processes resulting in demyelination and axonal loss, as well as the exhaustion of compensatory and repair mechanisms, lead to irreversible CNS damage [4]. Current disease-modifying therapies (DMTs) prevent disability progression and prolong MS patients’ lifetime, but MS remains incurable [5]. The most common MS phenotype is relapsing–remitting multiple sclerosis (RRMS) [6]. Disease activity is clinically expressed through relapses and neurological status progression, assessed by the Expanded Disability Status Scale (EDSS) [7]. The assessment of radiological activity in magnetic resonance imaging (MRI) plays a vital role in the monitoring of patients with MS. Radiological disease activity is determined by assessing the number, location, and volume of T2-weighted (T2-w) hyperintense lesions, as well as the presence of gadolinium-enhancing lesions (GELs). However, it is now well understood that MS involves not only focal inflammatory activity but also so-called smoldering inflammation, a chronic, ongoing inflammatory process that can manifest as the progression of brain atrophy [8].

Notably, brain atrophy emerged early as a sensitive and reliable predictor of future disability and is now increasingly used in clinical trials and practice to monitor disease progression and treatment response [9]. The most accurate method of atrophy estimation is a volumetric MRI evaluation of the brain tissues [10]. It does, however, require the acquisition of three-dimensional (3D) sequences, access to specialized software, and an experienced user [11]. Several feasible two-dimensional (2D) measurements of brain structures have been proposed to assess brain volume loss, with demonstrated high sensitivity, including frontal horn distance (FH), lateral ventricle width (LVW), intercaudate distance (CC), corpus callosum index (CCI), third ventricle width (TV), inner table of the skull measured along the CC line (IT), and inner table of the skull measured at its maximum width (mIT) [12].

The exact cause of MS is unknown, but vitamin D deficiency is one of the established risk factors [13]. Higher 25-hydroxyvitamin D (25(OH)D) serum levels across various racial and ethnic groups have been associated with a lower risk of MS [14]. Additionally, the beneficial effect of vitamin D intake on reducing this risk was confirmed in a large epidemiological study [15]. Currently, numerous genetic studies using Mendelian randomization have confirmed the association between lower serum 25(OH)D concentrations and a higher risk of MS [16,17].

The results from in vitro experiments, animal models, and analyses of clinical trial samples provide data indicating a pleiotropic effect of vitamin D on the immune system in MS patients [18]. This immunomodulatory effect is expressed through a reduction in the differentiation of effector T and B lymphocytes while promoting regulatory subgroups [19]. In addition, vitamin D inhibits the activation of microglia and astrocytes and stimulates oligodendrocyte proliferation, showing potential remyelinating and neuroprotective effects [20]. Vitamin D supplementation in MS patients was associated with an increase in anti-inflammatory cytokines, mainly interleukin-10 and transforming growth factor β (TGF-β), as well as an increase in regulatory cytokines such as interferon-γ, with less effect on pro-inflammatory cytokines such as interleukin-17 [21,22].

Many researchers have investigated the relationship between vitamin D serum levels and the clinical and radiological outcomes of disease progression. A meta-analysis involving thirteen studies with almost 3500 participants showed that an increase in 25(OH)D serum levels was associated with a reduction in clinical relapse rates, GELs, and new or enlarging T2-w lesions in MRI [23]. Similarly, a meta-analysis of fourteen studies (2817 MS patients) showed that an increase in serum 25(OH)D levels was associated with lower disability, as measured by the EDSS [24].

However, the results of randomized controlled trials (RCTs) evaluating the effect of vitamin D supplementation on disease progression remain inconclusive. Most authors did not demonstrate a therapeutic effect on the progression of disability measured by the EDSS and the incidence of relapses or annualized relapse rates [25,26,27,28]. In contrast, few meta-analyses have examined the effect on the radiological features of disease activity. A Cochrane meta-analysis did not find a therapeutic effect of vitamin D supplementation on the number of new GELs in MRI [29]. Analogous results were obtained regarding the presence of new T2-w lesions [30]. In contrast, a more recent meta-analysis showed the presence of statistical trends for new T2-w lesions, but these were not statistically significant [31]. Also, other authors noted that vitamin D supplementation appears to reduce the likelihood of new demyelinating lesions in MRI [32]. The current review of 35 observational studies—mainly prospective and retrospective cohort studies, but also cross-sectional studies—revealed that 60% showed a statistically significant relationship between vitamin D deficiency and radiographic activity [33].

Most studies have evaluated the effect of vitamin D supplementation on clinical activity (number of relapses and disability progression) as well as radiological activity (new T2-w lesions and GELs), yet very few examine the impact on brain atrophy. To address this issue, our study aims to assess the effect of vitamin D supplementation primarily on brain volume loss in MS patients.

## 2. Materials and Methods

### 2.1. Study Design

A total of 138 individuals treated in the Department of Neurology and the Outpatient Clinic of the Medical University of Silesia, Katowice, Poland, were recruited between October 2018 and April 2024. All participants provided informed consent to participate in the study. The patients were enrolled based on the following inclusion criteria:Diagnosed with MS (RRMS) according to the 2010 or 2017 revised McDonald criteria [34,35];Aged over 18 years;EDSS ≤ 6.5;Stable DMT treatment for at least one year at the baseline.

The exclusion criteria were as follows:5.Progressive subtypes of MS, including secondary progressive MS (SPMS) and primary progressive MS (PPMS);6.Relapse in the last 4 weeks;7.Steroid use in the last 6 weeks;8.An EDSS > 6.5;9.Pregnancy and breast-feeding;10.Acute or chronic renal failure;11.Hypercalcemia in medical history;12.DMT switch from platform to high-efficacy treatment agents or DMT discontinuation/termination;13.Other types of MS treatment (mitoxantrone and cyclophosphamide).

At the baseline, patients completed a standardized questionnaire designed to collect comprehensive data on demographic characteristics, factors influencing vitamin D status, and vitamin D supplementation. The first section of the questionnaire included variables such as age, sex, education level, height, weight, and skin phototype. The following section assessed sunlight exposure (daily duration, time of day, body surface area exposed, and sunscreen use), dietary habits (including the consumption of fatty fish, dairy products, eggs, and meat), comorbidities, smoking status, and corticosteroid use. Subsequently, all participants were asked about vitamin D supplementation. Those who reported supplement use provided additional details regarding the type of formulation, dosage, intake regimen, and adherence. Additionally, a comprehensive medical history was obtained, including MS-related information such as disease duration, disease activity, and previous and current treatments.

Effective sunlight exposure was defined according to the established criteria for optimal vitamin D synthesis, which included sunbathing for at least 15 min per day between 10:00 a.m. and 3:00 p.m. with exposure of the forearms and lower legs and without the use of sunscreen [36].

The study followed a prospective observational design, and no interventions were implemented as part of the study protocol. Data on vitamin D supplementation were collected, but no vitamin D was provided or prescribed within the study framework. However, during clinical visits, patients were informed about current recommendations regarding vitamin D supplementation.

Participants were prospectively followed for 48 months, with assessments conducted at five predefined time points. At every time point (baseline, 12 months, 24 months, 36 months, and 48 months), the following clinical and radiological outcomes were assessed: the number of new relapses, neurological status measured by the EDSS, number of new/enlarged T2-w lesions, and number of GELs in MRI. The 25(OH)D serum levels were also collected at every time point. A brain atrophy assessment was performed at the baseline and after 36 months of observation to ensure a measurable and clinically relevant interval between evaluations.

Further analyses were performed according to the vitamin D supplementation status. Follow-ups for sunlight exposure, lifestyle habits, and medical history were performed at every time point.

### 2.2. Atrophy Assessment

To assess the effectiveness of imaging parameters on the analysis of brain atrophy and the impact of vitamin D concentration, therapy, and other variables on its progression, multiple 2D measurements were performed and compared for all MS patients. Each patient’s annual MRI scans from 2018 to 2023 were individually assessed by an experienced neuroradiologist. The measurements included FH, CC, and TV, as well as IT and mIT, to calculate the BCR (CC/IT), Evans index (FH/mIT) [37], and frontal-horn–intercaudate-distance ratio (FH/CC). FH was defined as the distance measured between the lateral margins of the frontal horns at their maximum width. CC was defined as the distance measured between the caudate heads where they were closest. TV was measured at its maximum width. IT was measured between the inner table of the skull along the line of the CC measurement, whereas mIT was measured between the inner table of the skull at its maximum width. All MRI measurements were performed in the axial plane of a 3 mm T2-w sequence at the appropriate anatomical levels. The measurement methods are presented in Figure 1. The T2-w sequence was chosen as not all MRI examinations that were performed at different imaging diagnostic centers provided three-dimensional (3D) T1-weighted (T1-w) magnetization-prepared rapid gradient-echo (MPRAGE) sequences. Furthermore, the T2-w sequence allowed a more precise evaluation of IT and mIT widths without noticeable differences in other parameters compared with 3D T1-w MPRAGE or 3D fluid-attenuated inverse recovery.

### 2.3. Statistical Analysis

The statistical analysis was performed using STATISTICA 13.0 PL (Tibco Software Inc., Palo Alto, CA, USA) and R software (v 4.4.0; R Development Core Team (2008); R is a language and an environment for statistical computing from the R Foundation for Statistical Computing, Vienna, Austria). Statistical significance was set at a *p*-value below 0.05. All tests were two-tailed. Multivariate imputation by chained equations (MICE) was used to impute missing data. To impute univariate missing data, the predictive mean matching method was used (midastouch). Imputation was performed with the package ‘mice’, and based on 320 imputed datasets with 5 iterations in the predictive mean matching calculation. For all imputed datasets, a proper longitudinal analysis model was used and the results were pooled. Nominal and ordinal data were expressed as percentages, while interval data were expressed as the mean value ± standard deviation or 95% confidence interval (CI). Categorical variables were compared using χ^2^ or Fisher exact tests. Interval longitudinal data were analyzed with the mixed models for repeated measurements, with contrasts used for the post hoc analysis (package ‘mmrm’), while binary data were analyzed with a generalized linear model with a binomial function and logit link function. Multiple comparisons were corrected using the Hochberg method.

### 2.4. Ethics Approval

The study was conducted according to the guidelines of the Declaration of Helsinki and approved by the Ethics Committee of the Medical University of Silesia (No. KNW/0022/KB/135/19) extended in 2024 (No. BNW/NWN/0052/KB/31/24).

## 3. Results

### 3.1. Study Group Characteristics

Altogether, 132 patients were enrolled in the study (6 patients provided incomplete information regarding vitamin D supplementation). Of these, 13 patients discontinued the study at 24 months, another 10 patients at 36 months, and another 22 patients at 48 months. Patients were lost to follow-up or excluded for the following reasons: 23 patients did not have their vitamin D levels measured, 11 withdrew consent, 5 changed their treatment center, 5 became pregnant, and 1 patient died. At the baseline, the mean age of the studied group was 45.8 years (SD = 10.9). There were 98 women (74.2%) and 34 men (25.8%). The mean disease duration was 10.0 years (min: 6.0; max: 14.0). All patients were treated with the following DMTs, as presented in Table 1.

The study population was divided into two groups based on the vitamin D supplementation status. Of the participants, 97 patients received vitamin D supplementation and 35 did not. No statistically significant differences were found between the groups in terms of age (46.2 ± 10.9 vs. 44.7 ± 11.2 years; *p* = 0.48), BMI (25.2 ± 4.5 vs. 25.0 ± 4.7 kg/m^2^; *p* = 0.88), sun exposure (43.0% vs. 45.7%; *p* = 0.22), disease duration (10 [6; 14] vs. 9 [5; 13] years; *p* = 0.34), or HETA prevalence (16.5% vs. 14.3%; *p* = 0.76).

However, the proportion of male patients was significantly higher in the group without vitamin D supplementation (42.9% vs. 19.6%; *p* < 0.01). A higher prevalence of comorbidities and smoking was observed among supplemented patients (18.6% vs. 5.7% (*p* = 0.10) and 20.6% vs. 14.3% (*p* = 0.41), respectively), although these differences were not statistically significant. The differences between analyzed groups are shown in Table 2.

Among the patients receiving vitamin D3 supplementation, the most common daily dose was 2000 IU, reported by 37 individuals (38.1%). This was closely followed by 4000 IU, taken by 35 patients (36.1%). Lower doses were less frequent: 1000 IU was used by 14 patients (14.4%) and 500 IU was used by 7 patients (7.2%). Only 4 patients (4.1%) reported taking doses exceeding 4000 IU per day. There were no adverse events of vitamin D supplementation reported. Moreover, among subjects supplementing vitamin D, there were no correlations between dosage and age, BMI, FH, CC, IT, mIT, or TV/BCR values. Moreover, vitamin D dosage did not differ by sex, smoking status, sun exposure, or comorbidities. For reference, the mean supplementation dose was 2603 IU ± 1329 IU.

The patients’ diet was not included in the analysis as dietary intake alone does not provide an adequate supply of vitamin D, regardless of whether the sources are natural (such as fish, eggs, or liver) or fortified food products [38].

### 3.2. Vitamin D Serum Levels

The mean vitamin D serum level at the baseline was 21.7 ng/mL in patients without vitamin D supplementation and 41.2 ng/mL in patients with supplementation. The differences in the mean vitamin D serum levels between both groups were statistically significant at the baseline, 12, and 24 months. A significant increase in vitamin D serum levels was observed in patients without supplementation during the 4-year observation period (a difference of 18.3 ng/mL), reaching optimal levels after 36 months and 48 months. In contrast, in patients with supplementation, the vitamin D serum levels remained stable at an optimal level throughout the study period (a difference of −0.2 ng/mL). The vitamin D levels among the analyzed groups are presented in Table 3.

### 3.3. EDSS Progression

The mean EDSS value at the baseline was 2.6 pts in patients without vitamin D supplementation and 2.5 pts in patients with supplementation. There were statistically significant (or showing a tendency to be statistically significant) differences in the EDSS values between the third (*p* = 0.07) and fourth (difference of 0.58 (95% CI: 0.27–0.90) pts; *p* < 0.001) measurements in comparison to the baseline values among the whole studied group in patients with supplementation (difference between 48 months and the baseline of 0.63 (95% CI: 0.28–0.98) pts; *p* < 0.001). No changes in the EDSS values were observed in the group without supplementation (*p* = 0.47). The changes in the EDSS are presented in Figure 2.

### 3.4. Vitamin D Supplementation and Clinical and Radiological Disease Activity

In our study, no statistically significant correlation was found between vitamin D supplementation and the number of new relapses, EDSS changes, and radiological activity, including the number of new/enlarged T2-w lesions and GELs, over the 4-year observation period.

### 3.5. Vitamin D Supplementation and Brain Atrophy

After 36 months of observation, a significant (or showing to be statistically significant) increase in FH (*p* = 0.068), CC (*p* < 0.05), TV (*p* < 0.05), Evans index (*p* = 0.071), CC-to-IT ratio (BCR) (*p* < 0.01), and FH-to-CC ratio (*p* < 0.05) values was observed in comparison to the baseline values. No changes were noted for IT and mIT values throughout the study period.

In patients with vitamin D supplementation, the mean FH values were significantly (*p* < 0.05) lower than in patients without supplementation at the baseline and after 36 months of observation. Also, these patients had lower mean CC values—however, only at the 36 month observation—than patients without supplementation (*p* < 0.05). Moreover, the increase in CC values over 36 months of observation was significant in the group without supplementation, whereas in patients with supplementation, no changes were noted. The increase in CC/IT values after 36 months of observation was significant only in patients without supplementation (*p* < 0.05). Furthermore, the TV values were significantly higher in patients without supplementation than in patients with supplementation at the baseline and after 36 months of observation (*p* < 0.05). According to the Evans index and FH/CC values, there were no statistically significant differences among the studied groups. The results are presented in Table 4.

### 3.6. Sun Exposure

Adequate sun exposure was reported by 37.1% of patients, whereas 62.9% did not meet the criteria. Subjects with sun exposure were nearly twice as likely to have adequate vitamin D concentrations (OR = 1.73; 95% CI: 1.20–2.49), with 71.4% (95% CI: 64.9–77.1%) achieving sufficient levels compared with 59.0% (95% CI: 53.7–64.0%) among those without sun exposure (*p* < 0.01). Additionally, there was a significant influence of sun exposure (*p* < 0.001) and time of follow-up (*p* < 0.01) on the EDSS values. In each measurement throughout follow-up, subjects with sun exposure had lower EDSS values. As sun exposure affects both vitamin D levels and disability scores, we considered it in the analysis. There was no difference in vitamin D dosage between subjects with and without sun exposure (2712 ± 1329 IU vs. 2545 ± 1335 IU; *p* = 0.56). Moreover, sun exposure did not influence vitamin D levels or EDSS values at any time point (all *p*-values > 0.05). In addition, in the mixed-model analysis, including sun exposure as a factor did not affect vitamin D levels or EDSS scores.

## 4. Discussion

To the best of our knowledge, this is one of the first studies to examine the impact of vitamin D supplementation on brain volume loss in MS patients. This study reveals the possible protective role of vitamin D supplementation on brain volume, and confirms no association between supplementation and disease activity outcomes such as relapses, EDSS progression, and radiological activity.

### 4.1. Vitamin D Supplementation and Clinical and Radiological Disease Activity

In our study, vitamin D supplementation did not influence the clinical and radiological outcomes of disease activity in MS patients. Such a therapeutic effect of vitamin D supplementation has been observed by many authors [39,40,41], while others have shown a lack of a statistically significant relationship [42,43]. However, our results are consistent with the current meta-analyses by Hanaei et al. and Yuan et al. Mahler et al. emphasize the trends toward a reduction in new T2-w lesions in patients with vitamin D supplementation. The slight impact of vitamin D implementation on a reduction in radiological activity was reported in a previous study, but only in patients treated with a platform therapy [44]. It is our opinion that the lack of, or very little, impact of vitamin D supplementation on these outcomes may be concealed by the effectiveness of administered DMTs. Vitamin D supplementation is always evaluated as an added therapy to DMTs. Current DMTs are immunomodulating or immunosuppressive agents with a significant impact on inflammation in the CNS, whereas vitamin D supplementation may have only immunomodulating properties, particularly in the preservation of immune homeostasis and tolerance. Some authors point out the usage of higher doses of vitamin D and achieving higher than optimal serum levels (30–50 ng/mL) to demonstrate the potential effect of vitamin D supplementation on disease activity [45]. Currently, there is no evidence for these effects. Additionally, authors collectively point out the need for further studies with an adequate sample size to obtain reliable results that confirm or refute the effect of vitamin D supplementation on MS progression [46,47].

### 4.2. Vitamin D Supplementation and Brain Atrophy

In our study, brain atrophy progression—seen as increasing FH, CC, TV, Evans index, BCR, and FH-to-CC ratio values—was noticed in both groups of patients during the 36 months of observation. A gradual loss of brain volume is substantial in MS, approximately 0.5–1.35% per year, exceeding the rates of normal aging [48]. Brain atrophy proceeds at all stages of MS, as well as during treatment [49]. Both TV and FH showed significant differences between the patients with supplementation and patients without supplementation at the baseline and after 36 months of observation. According to the literature, TV and LV widths are useful measurements to evaluate whole-brain and grey matter volume and physical disability in MS [12]. In the course of MS, early TV enlargement was found to predict progressive disease and was associated with a higher risk of physical disability progression [50,51]. A significant difference in TV was also observed between RRMS and SPMS patients, with higher values reported for SPMS [50]. TV is also a good marker of whole-brain and grey matter volume and, along with the corpus callosum area, reflects the advanced global neurodegeneration process better than CCI in MS patients [50,51]. Our results are consistent with previous findings, although we chose a different method of TV measurement, assessing the widest portion of the third ventricle (usually located in the anterior part) instead of measuring it in the middle as reported in the cited studies [52].

Importantly, in patients with vitamin D supplementation, the mean FH, CC, and TV values were lower than in patients without supplementation at the baseline and after 36 months of observation. Additionally, the increase in CC and BCR (CC/IT) during the 36 months of observation was higher in the group without supplementation, suggesting the protective properties of vitamin D supplementation.

Previously, BCR was found to be useful in discriminating between MS patients with mild and severe brain volume loss, with the proposed cut-off value set at 0.162 [47]. Moreover, a lower BCR and a higher volume of T2-w lesions are more characteristic of MS than of systemic diseases with CNS involvement, allowing for a more accurate differentiation in uncertain cases [49]. CC, CCI, and TV were found to be sensitive markers of both disability progression and cognitive impairment in MS patients [50,51].

Similar to our results, Ascherio et al. demonstrated that a 50 nmol/L (20 ng/mL) increase in 25(OH)D was related to a 0.27% lower rate of brain loss as measured with the Structural Image Evaluation using Normalization of Atrophy (SIENA) method and focusing on the central cerebral volume [53]. Additionally, higher values of the 25(OH)D3-to-24,25(OH)2D3 ratio were associated with increased brain atrophy, defined as a lower brain parenchymal fraction (BPF) [54]. In the same study, lower levels of total 25(OH)D, 25(OH)D3, and 24,25(OH)2D3 were associated with higher Multiple Sclerosis Severity Scale scores, whereas lower levels of 24,25(OH)2D3 were associated with a higher EDSS. On the other hand, Fitzgerald et al. did not demonstrate any associations between 25(OH)D serum levels and brain volume, evaluated by normalized brain volume, during 12 months of observation [55]. Another study by Dörr et al. corresponded these findings and reported no significant differences regarding BPF in MS patients treated with high (20,400 UI) and low (400 UI) doses of vitamin D [46]. The last two studies did, however, cover short periods of 12 and 18 months, respectively.

The Evans index and FH/CC ratio did not prove to be good MS-specific brain atrophy markers in our study population. The latter most likely resulted from relatively proportional changes in both parameters, regardless of supplementation.

### 4.3. Potential Mechanism Underlying the Association Between Vitamin D Supplementation and Reduction in Brain Atrophy

Vitamin D supplementation may not significantly affect classical measures of disease activity, as DMTs play a key role in this context. However, it may have a beneficial impact on neurodegenerative processes and smoldering inflammation in MS, which are radiologically reflected by the progression of brain atrophy. The possible neuroprotective role of vitamin D has been evaluated by numerous studies in animal models of MS, including experimental autoimmune encephalomyelitis (EAE). The active form of vitamin D (calcitriol) reduces demyelination and increases remyelination in the CNS of EAE models [56,57,58]. Calcitriol also potentially supports cell survival in the CNS by increasing the expression of nerve growth factor in oligodendrocytes [59] and brain-derived neurotrophic factor in neural stem cells [60]. This could be the possible explanation for the correlation between vitamin D supplementation and reduced brain volume loss in MS patients, but this issue requires further investigation.

Furthermore, in another neurodegenerative disease, Alzheimer’s disease, a decrease in vitamin D serum levels was associated with a higher reduction in brain volume [61]. A similar correlation between vitamin D levels with focal brain atrophy was observed among patients with Parkinson’s disease [62].

Aging is increasingly recognized as a key factor contributing to neurodegeneration in multiple sclerosis. Age-related changes, such as immunosenescence, mitochondrial dysfunction, and reduced remyelination capacity, may exacerbate neuroaxonal loss and brain atrophy [63]. These processes may partly explain the progressive decline observed in our cohort, especially in older patients. Further research is warranted to disentangle the effects of aging from disease-specific mechanisms.

### 4.4. Consideration of Disability Progression

The observed EDSS increase in the fourth year of observation in the studied group is noteworthy. The time between specific EDSS levels varies considerably in the course of the disease. Disability accumulation intervals are shorter between EDSS levels 4 and 6 than between 0 to 3 [64]. It could also be the manifestation of disease transformation into a progressive phenotype or a result of shrinkage of the neurologic reserve. In the early stages of MS, symptoms may be shielded by an adequate neurologic reserve but, as brain atrophy proceeds, the neurologic reserve becomes scarce and patients enter the progressive stage of the disease [65]. We will explore this issue in the next retrospective analysis.

### 4.5. Sun Exposure Impact on Disability Progression

Our findings demonstrate that adequate sun exposure is associated with significantly higher rates of sufficient serum vitamin D concentrations among patients with MS, which reflects efficient cutaneous synthesis [66].

It should be noted that the observed association between sun exposure and slower disease progression may not be solely attributable to higher vitamin D levels. Other factors related to sun exposure, such as the immunomodulatory effects of UV radiation or behavioral and lifestyle aspects, may also contribute to this relationship [67]. Therefore, the causal pathway remains uncertain and warrants further investigation.

### 4.6. Study Strengths and Limitations

The main advantage of this study is its innovative topic, which addresses the investigation of the potential impact of vitamin D supplementation on brain volume loss in patients with MS. Another key strength is the long follow-up period, combined with the focus on brain volumetric outcomes, which are increasingly recognized as sensitive markers of disease progression.

However, some study limitations should be recognized. Firstly, the observational design precludes definitive conclusions about causality. Secondly, there is a potential for selection bias as the patients who agreed to participate and completed the follow-up could systematically differ from those who declined or dropped out, possibly affecting the generalizability of the results. Thirdly, the relatively high dropout rate over the 48-month follow-up period may have introduced attrition bias and reduced the statistical power of the analyses. Moreover, this study only examined patients with RRMS; progressive phenotypes were not included. Furthermore, the inclusion and exclusion criteria were quite broad and enabled the analysis of a standard population of patients with MS in clinical settings. We acknowledge the gender imbalance between the groups, which could be a potential confounding factor that may have influenced the results. As only 25.7% of the sample were men, we did not include gender in the analysis. Moreover, participants in the study took different amounts of vitamin D; however, our analysis did not specifically examine whether the dose had an impact on the outcomes. We agree that investigating a possible dose–response relationship could strengthen the conclusions. Further research is warranted to investigate the potential effects of varying the supplementation doses.

Additionally, the study was not designed as an RCT to assess the impact of vitamin D on disease progression due to the available guidelines of vitamin D supplementation for the general population and in patients with autoimmune diseases (including MS) [68]. All participants in the study were informed about these recommendations. This could have influenced the levels of 25(OH)D in the group without vitamin D supplementation, which increased by the end of the follow-up. Moreover, the evaluation of the impact of vitamin D supplementation was conducted only as an added therapy to DMT. To reduce the impact of DMT treatment on the study results, we decided to discontinue with the patients with a DMT switch from a platform to a HETA or with DMT discontinuation or termination. Finally, we focused on measurements in the transverse plane of the MRI scans, which are less demanding and quick to obtain in comparison to volumetric MRI evaluations. The atrophy assessment based on 2D measurements has been identified to correlate brain volume and clinical factors in MS; however, it is our opinion that the evaluation of the impact of vitamin D supplementation on atrophy progression assessed by volumetric parameters is crucial in MS patients.

## 5. Conclusions

Vitamin D supplementation among MS patients did not affect the number of new relapses or the progression of EDSS values, new/enlarged T2-w lesions, and GELs in our study. However, there was a statistically significant association between supplementation and a reduction in brain volume loss. Patients with vitamin D supplementation had lower atrophy rates as well as a smaller increase in brain atrophy after 36 months of observation than patients without vitamin D supplementation. FH, CC, BCR, and TV measurements were the most useful 2D parameters in the assessment of brain atrophy. These results were obtained from an observational study using clinical settings and should be interpreted as suggesting a possible benefit rather than implying a proven effect. The incorporation of brain atrophy measurements in RCTs to assess the impact of vitamin D on disease progression is warranted.

## Figures and Tables

**Figure 1 nutrients-17-02271-f001:**
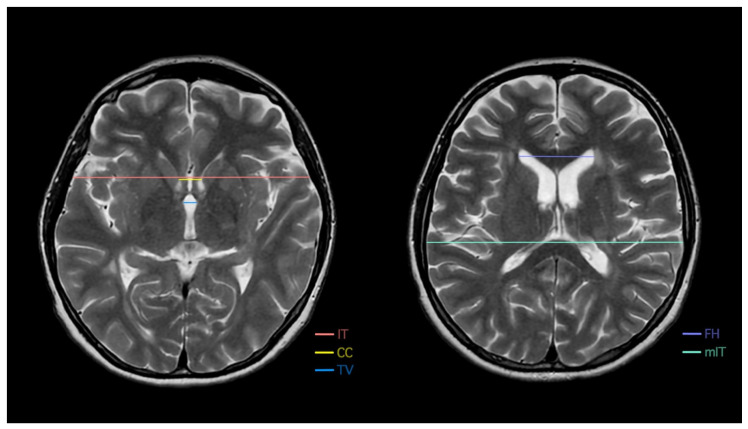
Two-dimensional (2D) methods of measuring brain atrophy parameters, including frontal horn distance (FH), intercaudate distance (CC), third ventricle width (TV), inner table of the skull measured along the CC line (IT), and inner table of the skull measured at its maximum width (mIT).

**Figure 2 nutrients-17-02271-f002:**
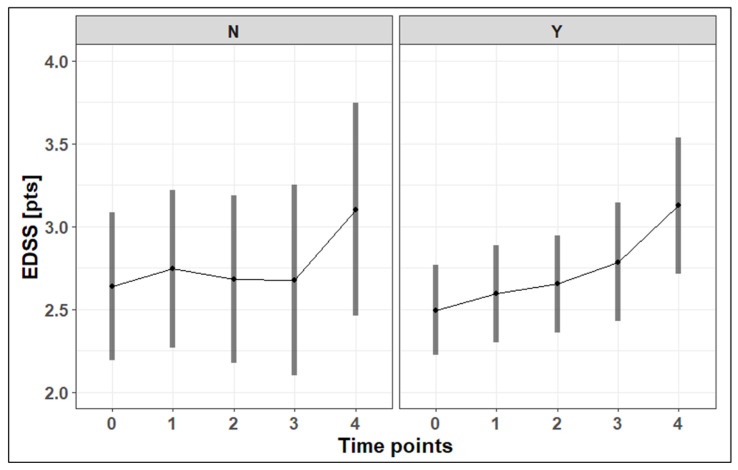
The changes in the EDSS among the studied group during follow-up. EDSS—Expanded Disability Status Scale; N—no supplementation; Y—supplementation.

**Table 1 nutrients-17-02271-t001:** Demographic and clinical characteristics of the studied group at the baseline.

Characteristics	Baseline (N = 132)
Men/women (n, %)	34 (25.8%)/98 (74.2%)
Age (mean ± SD)	45.8 ± 10.9
BMI (mean ± SD)	24.1 (21.9; 27.9)
Any comorbidities (n, %)	20 (15.2%)
Smoking (n, %)	25 (18.9%)
EDSS (median, min–max)	2 (1.5–6.5)
Disease duration (mean, min–max)	10.0 (6.0; 14.0) years
DMT (n, %)	132 (100%)
Interferons	21 (15.8%)
Glatiramer acetate	9 (6.8%)
Dimethyl fumarate	64 (48.5%)
Teriflunomide	17 (12.9%)
Fingolimod	11 (8.3%)
Natalizumab	8 (6.1%)
Ocrelizumab	1 (0.8%)
Alemtuzumab	1 (0.8%)
Platform/HETA	111/21 (84.1%/15.9%)

BMI—body mass index; EDSS—Expanded Disability Status Scale; DMT—disease-modifying therapy; HETA—high-efficacy treatment agent.

**Table 2 nutrients-17-02271-t002:** The baseline characteristics of analyzed groups (patients without vitamin D supplementation and with vitamin D supplementation).

	Patients Without Vitamin D Supplementation, N = 35	Patients with Vitamin D Supplementation, N = 97	*p*-Value
Male (n, %)	15 (42.9)	19 (19.6)	<0.01
Age (mean, ±SD) [years]	44.7 ± 11.2	46.2 ± 10.9	0.48
BMI (mean, ±SD) [kg/m^2^]	25.0 ± 4.7	25.2 ± 4.5	0.88
Sun exposure (n, %)	16 (45.7)	33 (43.0)	0.22
Comorbidities (n, %)	2 (5.7)	18 (18.6)	0.10
Smoking (n, %)	5 (14.3)	20 (20.6)	0.41
Duration of disease (median—lower quartile and upper quartile) [years]	9 (5; 13)	10 (6; 14)	0.34
HETA (n, %)	5 (14.3)	16 (16.5)	0.76

BMI—body mass index; HETA—high-efficacy treatment agent. Interval data are the mean ± standard deviation or median (lower quartile; upper quartile).

**Table 3 nutrients-17-02271-t003:** The mean vitamin D serum levels among the studied groups during follow-up.

	Patients Without Vitamin D Supplementation			Patients with Vitamin D Supplementation			
Follow-Up	Mean Value	Lower CI	Upper CI	Mean Value	Lower CI	Upper CI	*p*-Value
Baseline	21.7	13.9	29.4	41.2	36.5	45.9	<0.001
12 months	27.8	19.9	35.7	42.7	37.7	47.6	<0.01
24 months	28.9	20.9	37.0	42.5	38.0	47.1	<0.01
36 months	36.1	29.0	43.2	41.9	37.8	46.1	0.16
48 months	39.9	32.9	46.9	41.1	36.7	45.4	0.78
∆_48months_ vs. _baseline_	18.3	9.8	26.7	−0.2	−5.4	5.1	<0.001

CI—confidence interval.

**Table 4 nutrients-17-02271-t004:** The brain atrophy measurements at the baseline and after 36 months of observation among the studied groups.

Time Point	Patients Without Vitamin D Supplementation			Patients with Vitamin D Supplementation			*p*-Value
	Mean Value	Lower CI	Upper CI	Mean Value	Lower CI	Upper CI	
Frontal horn width (FH) [mm]							
Baseline	35.7	34.1	37.2	33.8	32.8	34.7	<0.05
At 36 months	36.3	34.4	38.1	34.2	33.1	35.2	<0.05
Intercaudate distance (CC) [mm]							
Baseline	12.0	11.0	13.0	11.1	10.5	11.7	0.13
At 36 months	12.7	11.6	13.8	11.4	10.8	12.0	<0.05
Inner table of the skull measured along the CC line (IT) [mm]							
Baseline	118	116	120	117	116	118	NS
At 36 months	118	116	120	117	116	119	NS
Inner table of the skull measured at its maximum width (mIT) [mm]							
Baseline	135.0	133.0	136.9	132.7	131.5	133.8	<0.05
At 36 months	134.5	132.5	136.5	133.3	132.1	134.4	0.29
Third ventricle (TV) [mm]							
Baseline	8.2	7.4	9.1	7.1	6.6	7.6	<0.05
At 36 months	8.7	7.8	9.6	7.4	6.8	7.9	<0.05
Bicaudate ratio (BCR)							
Baseline	0.100	0.092	0.108	0.095	0.090	0.100	NS
At 36 months	0.106	0.098	0.114	0.097	0.092	0.102	NS

CI—confidence interval; NS—non-significant.

## Data Availability

The data presented in this study are not publicly available due to privacy restrictions.

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
