# Peer review of "Does Vitamin D Supplementation Slow Brain Volume Loss in Multiple Sclerosis? A 4-Year Observational Study"

_nutrients, 2025, doi:10.3390/nu17142271_

Round 1
Reviewer 1 Report
Comments and Suggestions for Authors
I'm glad to be part of the review process for this manuscript. Overall, the study titled 'Does Vitamin D Supplementation Slow Brain Volume Loss in Multiple Sclerosis? A 4-Year Observational Study' presents interesting findings. However, I have several specific comments and concerns that the authors should address before the manuscript can be considered for publication.
- This is an interesting study, but since it’s observational, it can’t really show cause and effect. It would be good if the authors made this clearer in the abstract and discussion, and mentioned that randomized trials are needed to confirm the findings.
- There’s a noticeable difference in gender between the groups, with more men in the non-supplemented group. This could influence the results, so it might be helpful to adjust for gender or at least discuss this as a possible confounding factor.
- People in the study took different amounts of vitamin D, but the analysis didn’t explore whether dose made a difference. It would strengthen the paper to look at whether higher doses were more effective.
- The use of 2D brain scans is understandable, but they’re less precise than 3D methods. It would be helpful if the authors acknowledged this more clearly and explained how reliable the measurements were.
- Quite a few patients dropped out over the course of the study, but we don’t get much detail on why. It would be useful to know if those dropouts could have affected the final results.
- It’s a bit surprising that EDSS scores increased in the supplemented group. This seems worth a closer look—maybe there’s an explanation or additional analysis that could help clarify this.
- Since sun exposure can affect both vitamin D levels and disability scores, it seems important to consider it in the analysis. Adjusting for sunlight or looking at it separately would improve the interpretation of the results.
- The tables are detailed, but adding a few simple graphs could really help readers see the changes over time more easily. A visual summary would make the results more accessible.
- The conclusion currently feels a little too confident. It might be better to use more cautious wording—maybe say the findings “suggest a possible benefit” rather than implying a proven effect.
- Some parts of the discussion feel a bit repetitive. Tightening up the writing and focusing on the key messages would make this section easier and more enjoyable to read.
Author Response
Manuscript ID#: nutrients-3739552
Manuscript title: Does Vitamin D Supplementation Slow Brain Volume Loss in Multiple Sclerosis? A 4-Year Observational Study
Response to Reviewer #1’s Comments
We would like to thank the Reviewer for their careful reading of our manuscript and for the constructive and insightful comments, which have helped us to improve our work. Below we provide our point-by-point responses. Additionally, all changes have been highlighted in the revised manuscript using Track Changes for transparency.
Comment 1: I'm glad to be part of the review process for this manuscript. Overall, the study titled 'Does Vitamin D Supplementation Slow Brain Volume Loss in Multiple Sclerosis? A 4-Year Observational Study' presents interesting findings. However, I have several specific comments and concerns that the authors should address before the manuscript can be considered for publication.
Respond 1: We thank the Reviewer for the positive overall assessment of our study and for the valuable comments. We have carefully addressed all concerns as detailed below and revised the manuscript accordingly.
Comment 2: This is an interesting study, but since it’s observational, it can’t really show cause and effect. It would be good if the authors made this clearer in the abstract and discussion and mentioned that randomized trials are needed to confirm the findings.
Respond 2: Thank you for this valuable observation. We agree that our observational design does not allow us to establish causality. Accordingly, we have revised the abstract and discussion to explicitly acknowledge this limitation and to emphasize the need for future randomized controlled trials to confirm our results.
We add a sentence in Abstract in Methods Section: A 4-year observational, non-interventional study design was applied to evaluate association between vitamin D supplementation and disease progression.
We also add a sentence in Abstract in Conclusions: Randomized controlled trials are required to definitively establish the impact of vitamin D supplementation on brain atrophy progression.
Additionally, there is already a there is already a sentence in the Conclusion that emphasizes the role of randomized clinical trials: Still, these results were obtained from an observational study in clinical settings, the incorporation of brain atrophy measurements in RCT assessing the impact of vitamin D on disease progression is warranted.
Comment 3: There’s a noticeable difference in gender between the groups, with more men in the non-supplemented group. This could influence the results, so it might be helpful to adjust for gender or at least discuss this as a possible confounding factor.
Respond 3: Thank you for this valuable comment. We acknowledge the gender imbalance between the groups. As only 25.7% of the sample were men (resulting in a small number of male participants), we decided not to include gender as a variable in the analysis. We have added a note in the Discussion section acknowledging that sex could be a potential confounding factor.
Comment 4: People in the study took different amounts of vitamin D, but the analysis didn’t explore whether dose made a difference. It would strengthen the paper to look at whether higher doses were more effective.
Respond 4: Thank you for this valuable comment. We acknowledge that participants took different amounts of vitamin D; however, our analysis did not specifically assess whether the dose affected the outcomes. We agree that investigating a possible dose–response relationship could strengthen the conclusions. We therefore checked this and found that, among subjects supplementing vitamin D, there were no correlations between dosage and age, BMI, FH, CC, IT, mIT, or TV/BCR values. Moreover, vitamin D dosage did not differ by sex, smoking status, sun exposure, or comorbidities. For reference, the mean supplementation dose was 2,603 IU ± 1,329 IU. This information has been added to the Results section. We have also noted this point as a limitation and recommend that future studies explore the impact of different supplementation doses.
Comment 5: The use of 2D brain scans is understandable, but they’re less precise than 3D methods. It would be helpful if the authors acknowledged this more clearly and explained how reliable the measurements were.
Respond 5: Thank you for this important comment. We acknowledge that the use of 2D brain scans is less precise than 3D volumetric methods. However, due to the retrospective nature of our study, only 2D MRI sequences were available for analysis.
We have already discussed this in Study Limitations sections: Finally, we focused on measurements in the transverse plane of the MRI scans, which are undemanding and quick to obtain in comparison to volumetric MRI evaluation. The atrophy assessment based on 2D measurements has been identified to correlate to brain volume and clinical factors in MS, however, it is our opinion, that the evaluation of vitamin D supplementation impact on atrophy progression assessed by volumetric parameters is substantial in MS patients.
Comment 6: Quite a few patients dropped out over the course of the study, but we don’t get much detail on why. It would be useful to know if those dropouts could have affected the final results.
Respond 6: Thank you for this important comment. We have added information on the reasons for patient dropouts in the Results section. We acknowledge that patient dropouts during the study period could have influenced the results. To minimize the potential bias caused by missing data, we used multivariate imputation by chained equations (MICE), as described in detail in the Statistical Analysis section. Given the intensive use of imputation with 320 datasets, it is very unlikely that dropouts affected the final results.
Comment 7: It’s a bit surprising that EDSS scores increased in the supplemented group. This seems worth a closer look—maybe there’s an explanation or additional analysis that could help clarify this.
Respond 7: Thank you for this insightful comment. We acknowledge this point and have already explored it in the Disussion section. We have structured the Discussion into distinct subsections to highlight key aspects and improve clarity. We distinguished a dedicated subsection entitled Considerations on Disability Progression to provide a interpretation of this finding.
Comment 8: Since sun exposure can affect both vitamin D levels and disability scores, it seems important to consider it in the analysis. Adjusting for sunlight or looking at it separately would improve the interpretation of the results.
Respond 8: Thank you for this valuable comment. We agree that sun exposure could potentially influence both vitamin D levels and disability scores. In our analysis, there was no difference in vitamin D dosage between subjects with and without sun exposure (2,712 ±â€¯1,329 IU vs. 2,545 ±â€¯1,335 IU; p = 0.56). Moreover, sun exposure did not influence vitamin D levels or EDSS values at any time point (all p-values > 0.05). In addition, in the mixed-models analysis, including sun exposure as a factor also did not affect vitamin D or EDSS levels.
Results for mixed models’ analysis – vitamin D levels:
Vitamin D |
df1 |
df2 |
F.ratio |
p.value |
time |
4 |
97.80 |
3.507 |
< 0.05 |
supplementation |
1 |
120.00 |
11.815 |
< 0.001 |
sun exposure |
1 |
119.53 |
2.091 |
0.15 |
time:supplementation |
4 |
106.54 |
3.558 |
< 0.01 |
time:sun exposure |
4 |
105.07 |
0.209 |
0.93 |
supplementation:sun exposure |
1 |
119.79 |
0.569 |
0.45 |
all interactions |
4 |
105.49 |
0.299 |
0.88 |
Comment 9: The tables are detailed but adding a few simple graphs could really help readers see the changes over time more easily. A visual summary would make the results more accessible.
Respond 9: We appreciate this valuable suggestion. While we did not add new graphs to the manuscript, we have included a graphical abstract that visually summarizes the study design and main outcomes. We believe this adequately addresses the need for a clear visual overview and complements the detailed tables presented in the results.
Comment 10: The conclusion currently feels a little too confident. It might be better to use more cautious wording—maybe say the findings “suggest a possible benefit” rather than implying a proven effect.
Respond 10: Thank you for this valuable comment. We agree that more cautious wording is appropriate given the observational nature of the study. We have revised the Conclusion section to use more tentative language, stating that the findings suggest a possible benefit of vitamin D supplementation on brain volume loss rather than implying a definitive effect.
Comment 11: Some parts of the discussion feel a bit repetitive. Tightening up the writing and focusing on the key messages would make this section easier and more enjoyable to read.
Respond 11: We thank the reviewer for this constructive comment. We have also divided the Discussion into subsections addressing specific aspects of our findings to improve clarity and readability:
4.1 Vitamin D supplementation and clinical and radiological disease activity
4.2. Vitamin D supplementation and brain atrophy
4.3. Potential mechanism underlying the association between vitamin D supplementation and reduction of brain atrophy
4.4. Consideration on disability progression
4.5. Sun exposure impact on disability progression
4.6. Study Strengths and Limitations

Reviewer 2 Report
Comments and Suggestions for Authors
The manuscript presents a comprehensive 4-year observational study exploring the impact of vitamin D supplementation on brain volume loss in relapsing-remitting multiple sclerosis (RRMS). The study addresses a relatively underexplored aspect—vitamin D’s effect on brain atrophy—by using two-dimensional MRI parameters. The manuscript is generally well-structured and detailed. However, some areas require refinement for clarity.
Here my suggesitions:
The abstract is informative but should include specific numerical results (e.g., percentage difference in atrophy) for clarity.; Avoid vague terms like "possible correlation"; use "observed association" unless statistical causality is demonstrated.
Improve the background by avoiding repetitive phrasing (e.g., “pleiotropic effects” mentioned multiple times).
Clarify the rationale: why is brain atrophy a particularly relevant endpoint in such a disease, and how does it compare to other markers?
Clarify and summarize findings on brain atrophy with effect sizes or mean differences between groups.
Include drop-out rates and reasons.
Avoid over-interpretation in describing trends as significant if p > 0.05.
Strengthen discussion of mechanisms. Also the concept of aging would be of interest to include about neurodegeneration (i.e. doi: 10.3390/ijms18122672.)
Provide a more detailed discussion of limitations, particularly the observational design, potential selection bias, and the relatively high dropout rate.
Avoid speculative language like “possible protective effect”; instead, use terms appropriate for observational designs such as "associated with reduced progression."
Suggest that randomized controlled trials incorporating volumetric atrophy measures are warranted.
Improve English and spelling.
Avoid redundancy.
Author Response
Manuscript ID#: nutrients-3739552
Manuscript title: Does Vitamin D Supplementation Slow Brain Volume Loss in Multiple Sclerosis? A 4-Year Observational Study
Response to Reviewer #2’ Comments
We would like to thank the Reviewer for their careful reading of our manuscript and for the constructive and insightful comments, which have helped us to improve our work. Below we provide our point-by-point responses. Additionally, all changes have been highlighted in the revised manuscript using Track Changes for transparency.
Comment 1: The manuscript presents a comprehensive 4-year observational study exploring the impact of vitamin D supplementation on brain volume loss in relapsing-remitting multiple sclerosis (RRMS). The study addresses a relatively underexplored aspect—vitamin D’s effect on brain atrophy—by using two-dimensional MRI parameters. The manuscript is generally well-structured and detailed. However, some areas require refinement for clarity.
Respond 1: We sincerely thank the Reviewer for positive and encouraging feedback. We appreciate the acknowledgement of our work’s novelty and structure. In response to the Reviewer’s recommendation, we have carefully revised the manuscript to improve clarity and readability throughout.
Comment 2: The abstract is informative but should include specific numerical results (e.g., percentage difference in atrophy) for clarity.; Avoid vague terms like "possible correlation"; use "observed association" unless statistical causality is demonstrated.
Respond 2: We have revised the abstract to include key numerical results and replaced vague wording such as “possible correlation” with “observed association” to more accurately reflect the observational nature of the study.
Comment 3: Improve the background by avoiding repetitive phrasing (e.g., “pleiotropic effects” mentioned multiple times).
Respond 3: Thank you for this helpful observation. We have carefully revised the background section to eliminate repetitive phrasing and redundant expressions.
Comment 4: Clarify the rationale: why is brain atrophy a particularly relevant endpoint in such a disease, and how does it compare to other markers?
Respond 4: Thank you for this comment. We have emphasized this point more clearly in the Introduction to highlight why brain atrophy is an important and complementary endpoint compared to other markers. The classical radiological activity of the disease is determined by assessing the number, location and volume of T2-weighted (T2-w) hyperintense lesions, and presence of gadolinium-enhancing lesions (GELs). However, it is now well understood that MS involves not only focal inflammatory activity but also so-called smouldering inflammation — a chronic, ongoing inflammatory process that can manifest as the progression of brain atrophy. Notably, brain atrophy emerging early as a sensitive and reliable predictor of future disability and now increasingly used in clinical trials and practice to monitor disease progression and treatment response.
Comment 5: Clarify and summarize findings on brain atrophy with effect sizes or mean differences between groups.
Respond 5: Thank you for this comment. As we used the imputation method as well as mixed models’ regression such summary is not available.
Comment 6: Include drop-out rates and reasons.
Respond 6: We appreciate this comment. Dropout rates and reasons were included in the Results section, under the subsection Study Group Characteristics.
Comment 7: Avoid over-interpretation in describing trends as significant if p > 0.05.
Respond 7: Thank you for this important comment. We have carefully reviewed the manuscript to ensure that any trends with p-values greater than 0.05 are not described as statistically significant.
Comment 8: Strengthen discussion of mechanisms. Also the concept of aging would be of interest to include about neurodegeneration (i.e. doi: 10.3390/ijms18122672.)
Respond 8: Thank you for this important comment. We divided the Discussion section into subsections as follow:
4.1 Vitamin D supplementation and clinical and radiological disease activity
4.2. Vitamin D supplementation and brain atrophy
4.3. Potential mechanism underlying the association between vitamin D supplementation and reduction of brain atrophy
4.4. Consideration on disability progression
4.5. Sun exposure impact on disability progression
4.6. Study Strengths and Limitations
We have strengthened the discussion of potential mechanisms underlying our findings and expanded it to include the concept of aging as an important factor contributing to neurodegeneration in multiple sclerosis. In this context, we have incorporated the suggested reference (doi: 10.3390/ijms18122672).”
Comment 9: Provide a more detailed discussion of limitations, particularly the observational design, potential selection bias, and the relatively high dropout rate.
Respond 10: Thank you for this valuable comment. We divided the Discussion section into subsections and substantially expanded the Study Limitations subsection to provide a more detailed discussion of limitations, particularly addressing the observational design, potential selection bias, and the relatively high dropout rate.
Comment 10: Avoid speculative language like “possible protective effect”; instead, use terms appropriate for observational designs such as "associated with reduced progression.
Respond 10: Thank you for this valuable comment. We agree that using speculative language may overstate the findings in an observational design. We have revised the manuscript to replace phrases like “possible protective effect” with more appropriate wording such as “associated with reduced progression” to better reflect the nature of the data.
Comment 11: Suggest that randomized controlled trials incorporating volumetric atrophy measures are warranted.
Respond 11: We appreciate this comment. A statement indicating that randomized controlled trials with volumetric atrophy measures are needed is already included in the Conclusions section: Still, these results were obtained from an observational study in clinical settings, the incorporation of brain atrophy measurements in RCT assessing the impact of vitamin D on disease progression is warranted.
Comment 12: Improve English and spelling.
Respond 12: Thank you for your comment. We have carefully revised the manuscript to improve the English language and correct any spelling errors.
Comment 13: Avoid redundancy.
Respond 13: Thank you for your suggestion. We have removed redundant expressions to make the manuscript clearer and more concise.

Round 2
Reviewer 2 Report
Comments and Suggestions for Authors
The manuscript has been properly improved.